# Optimization of satellite-based communication links

**Bilal Muhammad Khan**[iD]◉*, **Muhammad Raheel**[iD]◉

National University of Sciences and Technology (NUST), Islamabad, Pakistan

◉ These authors contributed equally to this work.
* bmkhan@pnec.nust.edu.pk

## Abstract

Satellite communication (SATCOM) is crucial for global connectivity, particularly in remote far-flung areas. Typically, these links are established by connecting a Satcom Remote Terminal to a Satcom Hub Station. However, relying on a single Hub Station does not offer redundancy, leaving remote terminals isolated in case of a Hub Station failure. To provide redundancy, the remote terminals are shifted to a secondary Hub Station. However, several limitations and challenges are associated with the available methodologies for shifting of remotes to the secondary Hub Station. This research addresses the challenges associated with the implementation of redundancy/ Disaster Recovery (DR) mode in a Satcom Network to improve reliability and efficiency of Satcom links. An optimized model has been developed for Satcom Remote Modems to automatically switch to a secondary Hub Station during a primary Hub Station failure. The technique was tested in a real Satellite Communication Network, showing improved results compared to traditional methods.

## Introduction

Satellite communication has revolutionized global connectivity [1–3]. Communication satellites operate in various orbits, including geostationary, medium and low Earth orbits, enabling a wide range of communication services [4–8]. Roughly around 2,000 artificial satellites, orbiting Earth, ensure comprehensive coverage across the globe, making it possible to communicate seamlessly across continents and oceans [9,10]. Due to its global reach, satellite communication plays a crucial role in global telecommunication networks, making it essential to enhance their reliability and stability [11].

Satellite-based communication links are established at a remote site by connecting a Satcom Remote Terminal with a Satcom Hub Station through a communication satellite [12–14]. In a satellite-based communication link, a signal is transmitted from a Satcom Ground Station to a satellite. The satellite receives and amplifies the signal, then re-transmits the amplified signal back to Earth. Ground Stations receive and amplify the re-transmitted signal, facilitating useful communication [4,15,16].

Optimization of these satellite-based communication links is critical in enhancing the reliability and performance of global telecommunications networks. The issue has remained a part of latest advancements and ongoing challenges in satellite communications [17–20].

**Data availability statement:** The connectivity diagram / graph that are attached as Figures are generated by a software called "Ping Plotter". It directly plots the connectivity status in the form of a graph and provides traceroute while data sets of the graphs are not provided by the software. Since all the results are taken from a real time practical Hardware Setup. Moreover the software is an open source but proprietary software. It provides us with real time as well as past connectivity plot / graph but does not share information in the form of data sets. The software website as per journal policy is: https://www.pingplotter.com/

**Funding:** The author(s) received no specific funding for this work.

**Competing interests:** The authors have declared that no competing interests exist.

Several authors used Markov Bayesian Networks, Multi-objective Particle Swarm Optimization (MOPSO), and several other methodologies to model the reliability of satellite communication systems, allowing for systematic analysis and optimization of redundancy strategies [11,21–23]. Multiple design strategies have been proposed by the researchers to implement redundancy in Satcom links. One of the methodologies is to use multiple satellite paths for communication can enhance redundancy [24]. The optimization of satellite network infrastructure by focusing on gateway placement and routing is discussed in [25]. The optimization of frequency reuse in multibeam satellite systems to enhance capacity and reduce interference is explored by multiple researchers [26–30].

The two principal components that define the reliability of a satellite communication link are; the satellite and the ground station (Hub Station) [31,32]. Whereas, the reliability is significantly influenced by the ground stations that receive and transmit signals.

## SATCOM hub station redundancy

As far as, hardware is concerned, one of the key elements of Satcom network is the Hub Station. Satcom Hub Stations serve as pivotal points in Satcom networks, managing the transmission and reception of signals between satellites and end-users. Therefore, ensuring the availability of standby Hub Stations is vital for maintaining uninterrupted communication during disruptions [33].

Typically, a Satcom Remote Terminal is connected to only one Satcom Hub Station. Whereby, in the event of malfunctioning of the Hub Station, its corresponding remote terminals become standalone with no Satcom link [34,35]. In order to mitigate the situation, satellite communication providers install more than one Hub Stations [35–37]. However, these multiple Hub Stations does not provide any redundancy to the remote terminals connected with either of the Hub Stations until the remote terminals are shifted to the other Hub Station [38]. Therefore, a mechanism or methodology is implemented for shifting Satcom Remote Terminals from the parent Hub Station to a secondary Hub Station. This shifting procedure can be manual or automatic. This procedure for shifting communication links from primary to a standby Hub Station must be efficient and robust to minimize downtime and data loss [39].

### Manual shifting

In manual shifting, human intervention is required in which operators at the Satcom Hub Stations or Remote Terminals manually shift the remote users by re-configuring the remote terminals to connect to the secondary Hub Station.

### Automatic shifting

In this methodology, the Satcom Network is set up with pre-defined rules and conditions that trigger the automatic shifting of remote terminals. These rules can be based on criteria like signal quality, network congestion, or Hub Station failure. A Network Monitoring System (NMS) continuously assesses the status and performance of the parent Hub Station and Remote Terminals. When a trigger condition is met, the system automatically updates the configuration of the Remote Terminals to connect to the standby Hub Station without user intervention. 66

Here, we have discussed a generic manual and automatic methodology for shifting of 67 Satcom links from one Hub Station to the other. However, the specific mechanism for 68 shifting Satcom Remote Terminals from the parent Hub Station to a secondary Hub Station may vary depending on the vendor.

## iDirect SATCOM network

ST Engineering iDirect is a proprietary Satellite Communication technology. It is a prominent provider of the SATCOM equipment and is widely used in Pakistan across various sectors, including telecommunications, government, and enterprise. In the following paragraphs, Hub Station redundancy mechanisms used in an iDirect Satcom Network are explained: -

## Hub station redundancy in an iDirect network

In an iDirect Satcom Network, when two or more Hub Stations are deployed in a Satcom Network, the Hub Stations generally operate in Active-Active (Standalone) Mode, i.e., they do not provide any redundancy to the remote terminals connected with either of the two Hub Stations. In order to achieve redundant satellite communication links, the terminals need to switch between different networks, controlled from different Satcom Hub Stations which poses various challenges for IP Networks and Network Management Systems (NMS). This switching can be carried out manually as well as automatically [40,41].

## Manual shifting

When two iDirect Hub Stations are operating in Active-Active Mode, with standby arrangements in passive mode, following two methods are utilized for manual shifting of Satcom link of Satcom Remote Terminals from one HS to the other HS: -

1. Shifting during Routine Ops. During routine shifting of operations from parent HS to a secondary HS, when HS and its NMS are in serviceable state, manual shifting of remotes is carried out remotely through parent HS NMS by an operator. The procedure usually takes **15 Minutes** for shifting of Satcom link of one remote site. Total time taken for shifting of complete HS operations is the multiple of 15 Min with the total number of Satcom Remote Terminal connected with the HS.

2. Shifting during Disaster/ Contingency Situation. In case of any contingency situation in which parent HS or its NMS becomes unavailable, shifting of satcom link to secondary HS is carried out manually by remote site's staff himself. This manual shifting by site staff requires considerable time (**30 Minutes**) for shifting of operations. In addition, the procedure necessitates training as well as availability of requisite software (iDirect iSite), procedures, management terminal (laptop) and configuration file of secondary HS at all Satcom Remote sites.

## Automatic shifting

In order to implement redundancy/ Disaster Recovery (DR) mode, iDirect Global NMS is deployed in iDirect Satcom Networks to enable automatic shifting of Satcom Links between Hub Stations. Using this method, in the event of malfunctioning of primary HS, the Remote Modem tries to connect to primary HS for up to a defined time (five

minutes by default). Upon failed acquisition of primary carrier, the Modem attempts to connects to the secondary HS. The implementation requires significant amount of investment along with implementing architectural changes in the network connectivity and installation of additional hardware. Shifting of remotes through this automatic mechanism takes **6 - 8 Minutes**.

Summary of the available methodologies that can be used for shifting a Satcom Remote Terminal from one HS to another HS in an iDirect Satcom Network is tabulated in Table 1.

**Table 1. Summary of available shifting mechanisms.**

| S No | Description | Type | Time Req for Shifting | Other Additional Requirements |
|---|---|---|---|---|
| 1. | Shifting during Routine Ops | Manual | 15 Min | HS NMS & Operator |
| 2. | Shifting during Contingency | Manual | 30 Min | Management Terminal along with necessary files, SWs & Operator at Remote Site |
| 3. | Shifting with Global NMS | Automatic | 8 Min | iDirect Global NMS, additional HW & changes in NW |

### Limitations in shifting mechanisms

It can be concluded that several methodologies are available for shifting Satcom Links from one Hub Station to another Hub Station in an iDirect Satcom Network. The available procedures for manual shifting do not ensure redundancy or Disaster Recovery (DR) mode, thereby compromising the reliability of the Satcom Links. However, the available automatic shifting mechanism provides redundancy but requires the implementation of additional hardware and architectural changes.

Therefore, a need exists to develop an optimized shifting mechanism that can be employed without utilizing the iDirect Global NMS and without making major hardware or software changes to the network.

### Optimized shifting model

In order to optimize the Satcom connectivity of remote sites, several features of iDirect Satcom Network were studied during the research. The study included features of iDirect Satcom Remote Modems, Global & Conventional NMS, methodology used in iDirect Global NMS for implementation of redundancy/ DR mode and procedure for a Satcom Modem to register on an iDirect network.

### Registration on an iDirect HS network

During the research, it is ascertained that a Satcom Remote Modem registers on an iDirect HS Network and establishes Satcom link based on the following parameters: -

- Downstream Frequency Carrier (Tx Carrier of HS)

- DID (Derived Identification) of the Satcom Modem

- IP Configuration of the Satcom Modem

All this information is uploaded/ configured on an iDirect Satcom Modem through an Option File. An option file is a critical configuration file used to set various parameters necessary for the router to establish and maintain a connection with the Hub Station Network [42]. This file contains a range of settings, including frequency assignments, network IDs and other operational parameters. If an additional DS frequency carrier, i.e., DS Carrier of the standby HS can be configured on a Satcom Modem and the remaining two parameters of the Modem can be made unique for both the primary and secondary Hub Stations, the Modem can be registered on either of the Hub Stations.

### Alternate Carrier

iDirect iBuilder software, used for configuration of iDirect HS network, provides an option to configure a frequency carrier as an alternate downstream carrier on the HS Network and

Remote Modems. In case the primary carrier becomes unavailable, remote modem searches the primary carrier for 05 minutes (by default) and shifts to the alternate carrier automatically after stated time period [43,44].

### DID (Derived Identification)

A unique DID for a Satcom Modem is derived from its Serial No for registration on HS Network. The DID is a part of its configuration file [42]. During the research, it is ascertained that the DID of a modem remains same for different HSs if the iDirect Evolution Software Version of the HSs is same. Therefore, if both the HSs are operating on same software versions, same DID is derived for a Satcom Modem on both HSs. Thus, making it possible for the modem to register with a secondary/ standby HS while using the configuration file of its parent HS.

Same methodology can be utilized to produce one Option file for a Satcom Modem with an Identical DID that will be acceptable at both the primary and secondary/ standby Hub Station for registration of the Modem on the HS Network.

### IP configuration

The IP configuration of a Satcom Modem is critical for its successful registration and operation with a HS network. Proper IP configuration ensures that the modem can communicate effectively with other network components, manage data traffic and provide reliable service to end-users. Similarly, in an iDirect Satcom HS Network, IP configuration is crucial for registration over the network. It includes: -

- **IP Address**. The IP address assigned to the Modem.

- **Subnet Mask**. Defines the network segment used by the Modem.

In order to establish Satcom link with a Satcom HS, IP configuration (IP Address and User VLAN IPs/ Subnet Mask) of a Satcom Remote Modem should be configured same on both the parent and secondary HSs. Since, HSs operate in separate isolated LANs; therefore, same IP addresses and VLANs can be configured for a Satcom Modem on both primary and secondary/ standby Hub Station HSs without causing IP conflicts allowing for registration of the Modem on both the Networks.

### Unique option file

Keeping in view the above-stated parameters, a Unique Option File has been developed for registration of a Satcom Modem on both the primary and secondary HSs with same Option File. The procedure will be helpful in implementing redundancy/ DR mode in a Satcom Network through automatic shifting of Satcom link from primary HS to a secondary/ standby HS without deploying iDirect Global NMS. Key features of the developed solution are: -

- **Dual Carrier Configuration.** The Option File will contain the downstream carrier frequency information of both the primary HS and the secondary or standby HS.

- **Unique DID**. It will carry the DID that will be applicable for registration on both the primary and the secondary HSs.

- **IP Configuration**. IP configuration will be the same and will facilitate registration of Modem on both the primary and secondary networks.

- **Automatic Carrier Switching**. It includes a predefined timeout period during which the modem attempts to reacquire the primary carrier before switching to the secondary carrier.

After the timeout, if the primary carrier is not reacquired, the modem automatically shifts to the secondary carrier, ensuring continuity of service.

- **No Major Hardware or Software Changes**. The Option File is designed to work with existing hardware and software configurations, eliminating the need for iDirect Global NMS, any significant changes in network architecture or additional equipment.

## Proposed algorithm

The principles of the proposed algorithm rely on a proactive and automated switching mechanism between primary and secondary Hub Stations.In this section a detailed breakdown of how the unique option file leverages iDirect's technology and underlying principles:

### Automatic registration mechanism

The unique option file allows Satcom modems to automatically register with both the primary and secondary Hub Stations (HS) using predefined settings. This file configures the necessary parameters for modem registration, such as:

- **Downstream Frequency Carrier (DS Carrier):**This represents the frequency on which the modem communicates with the HS. The option file is designed to include both the primary and secondary DS carriers. When the primary HS fails, the modem searches for the secondary carrier and automatically switches to it after a predefined timeout period, ensuring uninterrupted service.

- **Derived Identification (DID):**The DID is a unique identifier for each modem, generated from its serial number. The algorithm optimizes this feature by ensuring that the DID remains consistent across both HSs if they operate on the same software version. This consistency enables the modem to automatically register on either HS without requiring different DID configurations, simplifying the process of redundancy.

### Alternate frequency carriers

One of the critical features of the iDirect platform is its ability to handle multiple frequency carriers, allowing the modem to switch to an alternate carrier when the primary carrier fails. The unique option file configures an alternate DS carrier on the modem, ensuring that when the primary carrier becomes unavailable, the modem will automatically search and switch to the alternate frequency carrier (from the secondary HS) after a set timeout. The algorithm leverages iDirect's iBuilder software, which provides the flexibility to configure multiple downstream carriers. By preconfiguring both carriers in the option file, the modem can switch seamlessly between the carriers based on network availability, without manual intervention. This capability drastically reduces downtime and eliminates the need for additional hardware changes like the installation of iDirect Global NMS.

### Building on frequency switching

The algorithm optimizes the frequency switching process by introducing a timeout mechanism. When the modem loses the primary DS frequency signal (i.e., when the primary HS is unavailable), it continues to search for this signal for a predefined period (default is 5 minutes). If the modem fails to reconnect to the primary frequency, it switches to the alternate frequency of the secondary HS. During the trials, the default timeout period was optimized to reduce the switchover time from 6 minutes 20 seconds to 3 minutes 20 seconds by adjusting

the search time for the primary carrier. This optimization balances the need to reconnect to the primary carrier while ensuring minimal communication disruptions by transitioning to the secondary carrier promptly.

### IP configuration

Another vital component is the modem's IP configuration, which is crucial for seamless communication with both HSs. The option file contains:

- **IP Address and Subnet Mask:**The same IP address and VLAN settings are configured for the modem across both the primary and secondary HSs, ensuring that the modem can register on either network without encountering IP conflicts. 252

- **VLAN Configuration:** By configuring the same IP and VLAN settings on both HSs, the modem can maintain uninterrupted data flow even after switching to the secondary HS. This setup ensures that the modem remains accessible and can communicate with the rest of the network as soon as it connects to the secondary HS.

The use of consistent IP and VLAN configurations in separate isolated LANs ensures that the modem can switch between networks without requiring significant changes to the network architecture.

### Seamless transitions between hub stations

The algorithm builds on iDirect's existing capabilities to provide seamless transitions between HSs. By combining alternate carrier configuration, DID consistency, and identical IP configuration, the modem can detect primary HS failures and automatically initiate a switchover to the secondary HS, ensuring that communication is restored with minimal disruption. The automatic carrier switching and IP configuration settings in the option file allow this transition to occur without requiring major changes to the network hardware or architecture, making the process cost-efficient and effective. In
    summary, the proposed algorithm in the manuscript optimizes Satcom reliability by leveraging key features of iDirect Satcom technology:

- **Automatic registration:** using consistent DID across HSs.

- **Frequency switching:** facilitated by dual carrier configurations.

- **IP management:** that ensures smooth transitions between HSs without IP conflicts.

This method significantly enhances network resilience without introducing the need for additional infrastructure like the iDirect Global NMS. The Algorithm layout is depicted in the Fig 1.

### Description of algorithm

Each Satcom Remote Modem is pre-configured with connectivity details (i.e., Downstream Frequency Carrier) for both the primary and secondary Hub Stations using Unique Option File. If the primary Hub Station becomes unresponsive, i.e., its Downstream Frequency Carrier becomes unavailable, the algorithm starts a countdown timer. During this time, the modem attempts to reacquire the primary carrier. If the primary Hub does not recover within the set timeout, the algorithm triggers the switching process. Upon timeout, the algorithm automatically directs the Satcom Remote Modem to shift its connection to the secondary Hub Station by switching to secondary frequency carrier using the pre-configured parameters. This action is done without manual intervention, enabling rapid re-establishment of the Satcom link. Upon acquiring the Secondary frequency carrier, the Satcom Modem is registered with

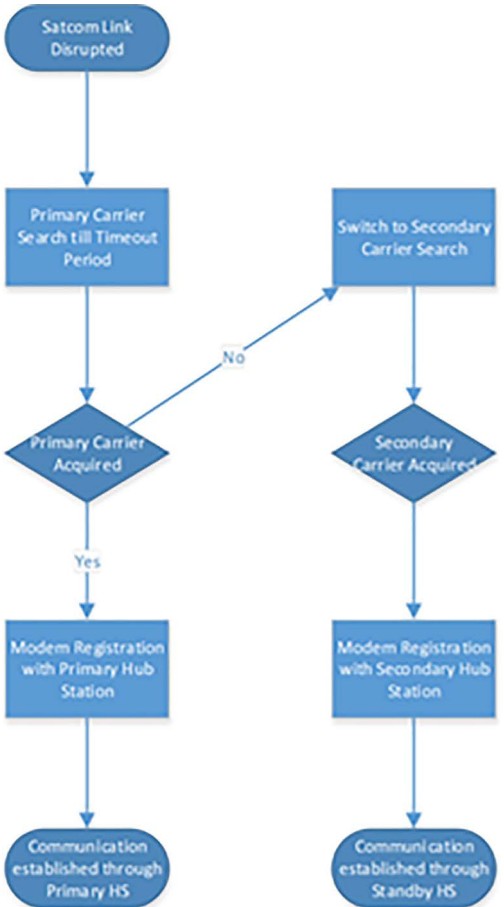

**Fig 1. SATCOM hub station redundancy algorithm.**

the secondary Hub Station and Satcom link is established through the secondary Hub Station to provide communication services at the remote site. Moreover, it is important 291 to mention that both the primary and secondary hub stations operate on the same iDirect Evolution Software version. This can be a limitation, however, since the algorithm is developed on running hardware actual system this limitation can be justified and further rectified in the following manner.

- **Software Version Compatibility:** In the proposed technique, the assumption of both hub stations running the same software version is based on the practical need for operational consistency across the network. The iDirect Evolution system, as per current implementation standards, requires the same software environment for seamless redundancy and disaster recovery (DR) operations. This ensures that the satellite modems can transition between the primary and secondary hubs without changing configurations or encountering operational discrepancies.

- **Mitigation of Version Mismatch Risks:** Software updates can cause version mismatches between hubs and pose challenges to the automatic switching mechanism. To mitigate this, the proposed technique implements a version management protocol where all hub stations are updated synchronously. This can be achieved through planned network-wide upgrades and rigorous pre-testing on a non-operational system before deployment in the

live network. Additionally, mechanisms such as backward compatibility checks during software updates ensure that the system maintains functionality even if there are minor version differences between hubs.

## Integrating proposed technique with existing satellite technologies

The proposed technique, centered around the Unique Option File and automatic registration mechanisms, enhances satellite communication networks, particularly those operating on the iDirect platform, by addressing key challenges in redundancy and downtime. Here's how this technique integrates with existing satellite communication infrastructures and how it is adaptable to more advanced systems:

### Enhancing iDirect-based Networks

The iDirect platform is widely used in satellite communications, offering flexibility and robust network management. However, iDirect systems often rely on manual or hardware-intensive solutions for switching between Hub Stations (HS) during failures, which can lead to communication delays and additional costs.

- **Automatic Hub Station Redundancy:** The proposed algorithm enhances iDirect networks by automating the switch between primary and secondary HSs without requiring iDirect's Global NMS or significant infrastructure changes. This automatic shifting minimizes the need for manual intervention, reduces downtime, and ensures continuous connectivity. The use of predefined DS carriers and consistent Derived Identification (DID) for modems ensures that iDirect systems can handle HS failures seamlessly, improving overall network resilience.

- **Optimized Switching Time:** By reducing the switching time from over 6 minutes to approximately 3 minutes, the technique provides faster failover, minimizing disruptions for users dependent on satellite links, especially in critical scenarios like disaster recovery, remote communications, or emergency services.

### Compatibility with common satellite communication infrastructures

Satellite networks are categorized into geostationary (GEO), medium Earth orbit (MEO), and low Earth orbit (LEO) systems, each presenting unique challenges in terms of latency, coverage, and mobility. The proposed technique is compatible with all these satellite architectures, enhancing reliability across different platforms.

- **GEO Satellites:** Geostationary satellites are used for long-term, stable connections due to their fixed positions relative to Earth. The proposed redundancy technique fits well in GEO satellite systems, as these networks often require reliable ground station handovers. The Unique Option File's ability to switch between Hub Stations ensures continued service in case of a GEO satellite's ground segment failure, making it highly beneficial for commercial and governmental services using GEO systems.

- **MEO and LEO Satellites:** MEO and LEO satellites, being closer to Earth, provide lower latency but involve more frequent handovers due to their movement relative to the ground. In such dynamic environments, the proposed algorithm can be adapted to enhance redundancy and smooth transitions between ground stations, ensuring that communication is not disrupted as satellites move in and out of range. The technique can be further optimized for LEO systems by adjusting the timeout periods for switching between Hub Stations to account for the more rapid movement and handover frequency in LEO networks.

- **Adaptive to Changing Orbits:** In MEO and LEO networks, where satellites frequently switch ground stations, the algorithm's use of automatic registration and alternate frequency carriers ensures a smoother transition. The modem's preconfigured settings allow for quick registration and failover, making it suitable for satellite constellations with frequent handovers.

## Adaptation to advanced communication systems (5G, future constellations)

As satellite communications continue to evolve, the integration of 5G and future satellite constellations such as mega-constellations (e.g., Starlink, OneWeb) will necessitate robust redundancy and seamless handover mechanisms. The proposed technique is well-positioned to integrate with such advanced systems due to its flexibility and reliance on existing network protocols without major hardware changes.

- **5G Integration:**In 5G-enabled satellite networks, where high-speed data transfer and low-latency communication are critical, the algorithm's ability to rapidly switch between Hub Stations ensures that 5G satellite backhaul services remain uninterrupted, even during ground station failures. The integration of automatic switching in 5G satellite infrastructure would provide continuous service during network disruptions, meeting the stringent requirements of 5G connectivity.

- **Support for Satellite Constellations:** Future satellite constellations, especially those operating in LEO, involve hundreds to thousands of satellites working together to provide global coverage. These constellations demand efficient handovers between satellites and ground stations. The proposed technique's compatibility with LEO systems can be extended to handle the dynamic handover requirements of these constellations by optimizing carrier switching times and ensuring that modems can quickly register with the most appropriate Hub Station, regardless of satellite movement.

- **Scalability:**The solution is scalable to handle larger networks, including mega-constellations, as the same principles of automatic registration, consistent DID, and frequency switching can be applied to more complex networks. This adaptability makes the technique future-proof as satellite communications increasingly converge with terrestrial networks and new technologies.

## Broader network integration

- **Hybrid Satellite-Terrestrial Networks:**As hybrid systems that combine satellite and terrestrial networks become more common, the algorithm can be adapted to manage failovers between satellite and terrestrial hubs. In these hybrid systems, when a terrestrial connection is lost, the modem could automatically switch to a satellite-based Hub Station, ensuring uninterrupted connectivity in remote or underserved areas.

- **Disaster Recovery and Resilience:**The technique also enhances network resilience in disaster recovery scenarios. For instance, in the event of a ground station failure due to natural disasters, the modem can quickly switch to a backup Hub Station, maintaining critical communications for first responders or humanitarian relief teams operating in affected areas.

The proposed technique enhances current satellite communication networks by integrating seamlessly with the iDirect platform and providing optimized automatic switching mechanisms. Its compatibility with various satellite infrastructures, from GEO to LEO, and its adaptability to advanced systems like 5G and future satellite constellations, ensures that the solution

is scalable and future-ready. By reducing downtime and minimizing the need for manual intervention or expensive hardware upgrades, the proposed solution significantly improves network resilience, making it a valuable enhancement for the rapidly evolving satellite communication landscape.

## Implementation & Results

### Technical arrangements for test/ trials

In order to validate automatic shifting of Satcom Remote Terminals from parent HS to a secondary HS by utilizing the Unique Option File, test/ trials were conducted on a real satellite communication link. Technical arrangements, made for implementing the proposed methodology on an iDirect Satcom Network, are described in the following sub-sections.

### iDirect satcom network

A satellite communication network consisting of two iDirect Satcom Hub Stations and a Satcom Remote Terminal was used to test the concept. One Hub Station served as the primary HS for the Satcom connectivity of the remote terminal, while the other Hub Station functioned as a secondary or backup HS. Both Hub Stations were connected to the WAN to provide the necessary communication services to the end-user via the Satcom link. In addition, a communication server, available on the WAN, was used as a Target Server to test connectivity through the Satcom link. The connectivity layout is depicted in the Fig 2.

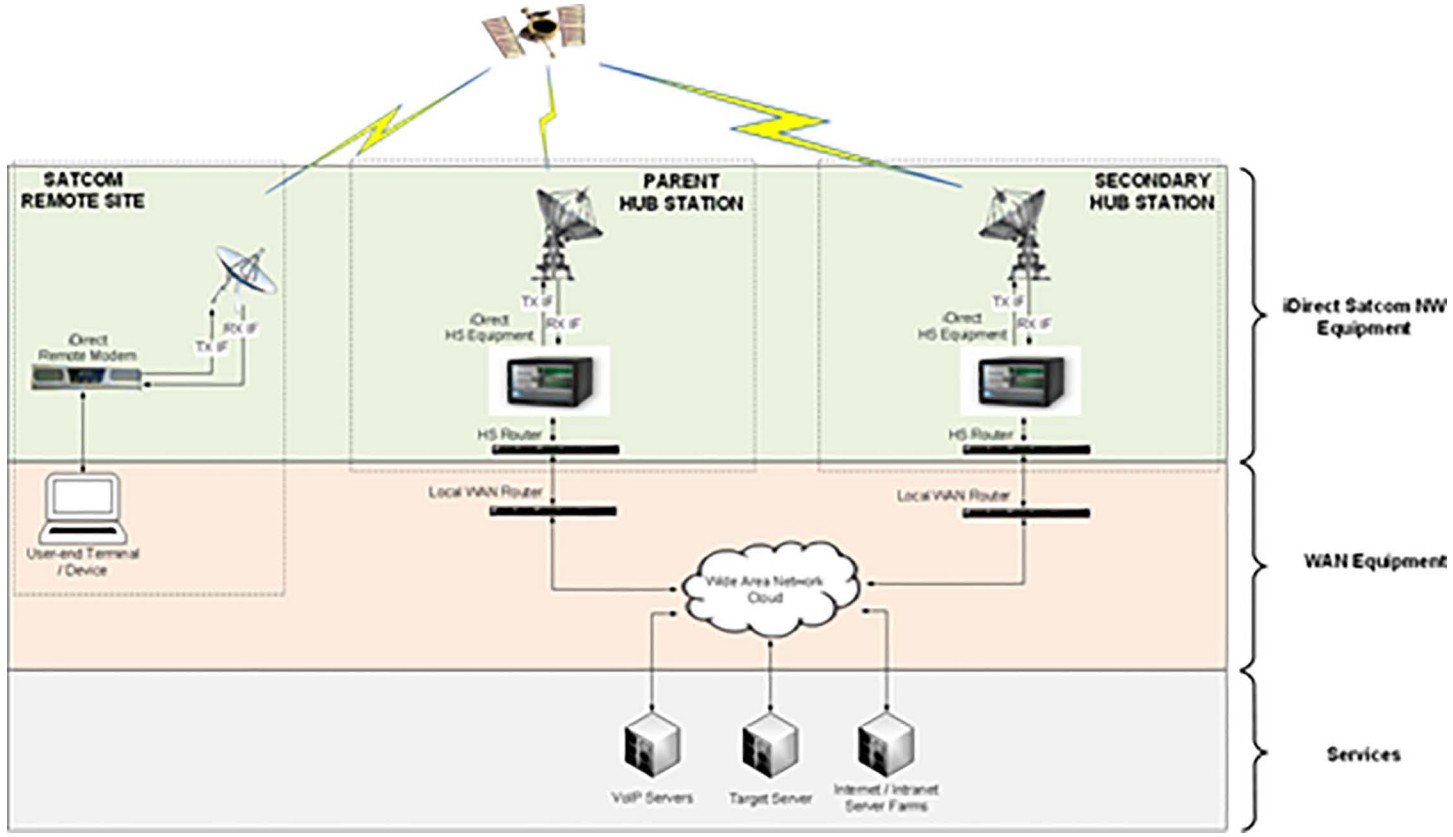

**Fig 2. iDirect satcom network used for test/ trials.**

### Generation of unique option file

A Unique Option File, designed to facilitate registration and operations of a Satcom Modem with both the primary and secondary HSs, was prepared in accordance with the proposed technique.

- **Modem Registration**. A Satcom Modem was registered with both the Satcom HSs. The replication of Derived IDs (DIDs) generated for the Modem at NMS of both the HSs was verified. This process ensured that the Modem can seamlessly switch between the two HSs while maintaining consistent identification.

- **Alternate Carrier Configuration**. Downstream carrier frequency of secondary HS was configured/ included in the Option File of the Modem as an alternate carrier. Refer Fig 3.

- **IP Configuration**. Same management IP address, subnet mask and user VLAN IPs were configured for the Modem at both HSs.

After configuring the parameters mentioned above, the Unique Option File was generated and uploaded onto the Satcom Modem and a Satcom link was established through the primary Satcom Hub Station.

## Test/ Trials

Subsequent to the necessary configuration and technical arrangements, testing for automatic shifting of the Remote Modem operations from parent HS to secondary HS was carried out successfully. The shifting of operations was verified by tracing route of the IP traffic of the Remote Modem towards a server (xx.xx.130.26) present on the Wide Area Network (WAN). Details of the testing are as follows: -

### Test/ trials with default timeout setting

In the first phase of testing, default timeout setting of 05 Minutes (300 Seconds) for searching primary DS frequency carrier before shifting to secondary DS carrier was used.

### Satcom Link with Parent HS

Connectivity of the remote modem with primary HS was verified. During operations on primary downstream carrier, i.e., when the remote terminal was connected with the primary HS, the traffic routed from primary HS Router (xx.xx.1.1) to local WAN Router (xx.xx.10.2) and then towards Target Server (xx.xx.130.26) in 07 hops over WAN. Fig 4 and Table 2 depicts that the primary DS carrier (Beam 73) is selected on the remote modem and the IP traffic route from user terminal to the Target Server while Satcom link was established through the primary HS.

### Shifting of satcom link from primary HS to secondary HS

Downstream frequency carrier of the primary HS was made unavailable by Switching 'off' the transmission of the primary HS. Resultantly, Satcom link of the Remote Modem became

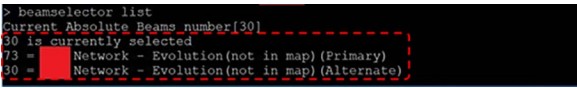

**Fig 3. Satcom Modem CLI showing Configuration of Primary & Alternate Carriers.**

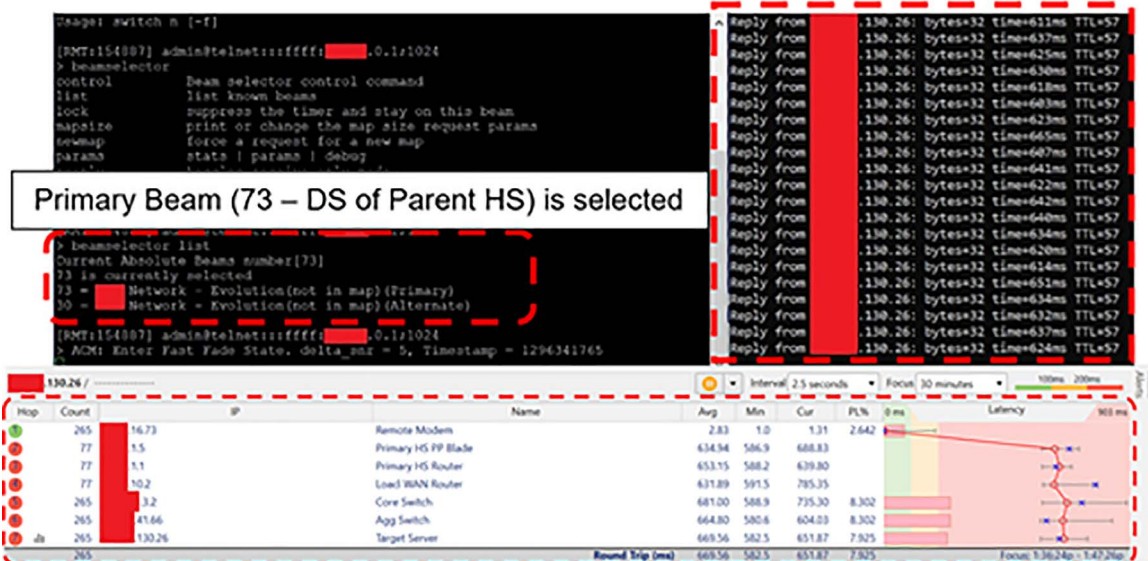

**Fig 4. Remote Modem IP Traffic routed through Parent HS.**

**Table 2. Traffic route while satcom link is through parent HS.**

| No of Hop | NW Element | NE IP Address |
|---|---|---|
| 1 | Remote Modem | xx.xx.16.73 |
| 2 | Primary HS PP Blade | xx.xx.1.5 |
| 3 | Primary HS Router | xx.xx.1.1 |
| 4 | Local WAN Router | xx.xx.10.2 |
| 5 | Core Switch | xx.xx.3.2 |
| 6 | Aggregation Switch | xx.xx.41.66 |
| 7 | Target Server | xx.xx.130.26 |

unavailable and shifted to the alternate carrier (DS carrier of secondary HS). Upon secondary carrier acquisition, the Remote Modem was successfully registered with the secondary HS.

The Remote Modem took 06 Minute and 20 Seconds to establish Satcom Link with the secondary HS. 05 Minutes for searching Primary Carrier before shifting to the Alternate Carrier were taken as per default settings. In addition, 01 Minute and 20 Sec were taken to acquire DS carrier of secondary HS, registering with the secondary HS and establishing the communication link. Thus, time taken for complete shifting cycle from parent HS to a secondary HS with default timeout settings was 06 Minutes & 20 Seconds [5 Min (for Carrier Search) + 01 Min & 20 Sec (for Network Acquisition/ Registration)] Fig 5 presents results of test/ trials and depicts the time taken by the Remote Modem to shift from primary to secondary HS.

## Satcom link with secondary HS

Subsequently, connectivity of the Remote Modem with secondary HS was verified. During operations on alternate downstream carrier, i.e., secondary HS, the traffic was routed from secondary HS Router (xx.xx.13.1) to local WAN Router (xx.xx.20.2) and then towards the Target Server (xx.xx.130.26) in 11 hops over WAN Cloud. Fig 6 and Table 3 depicts that the secondary DS carrier (Beam 30) is activated on the

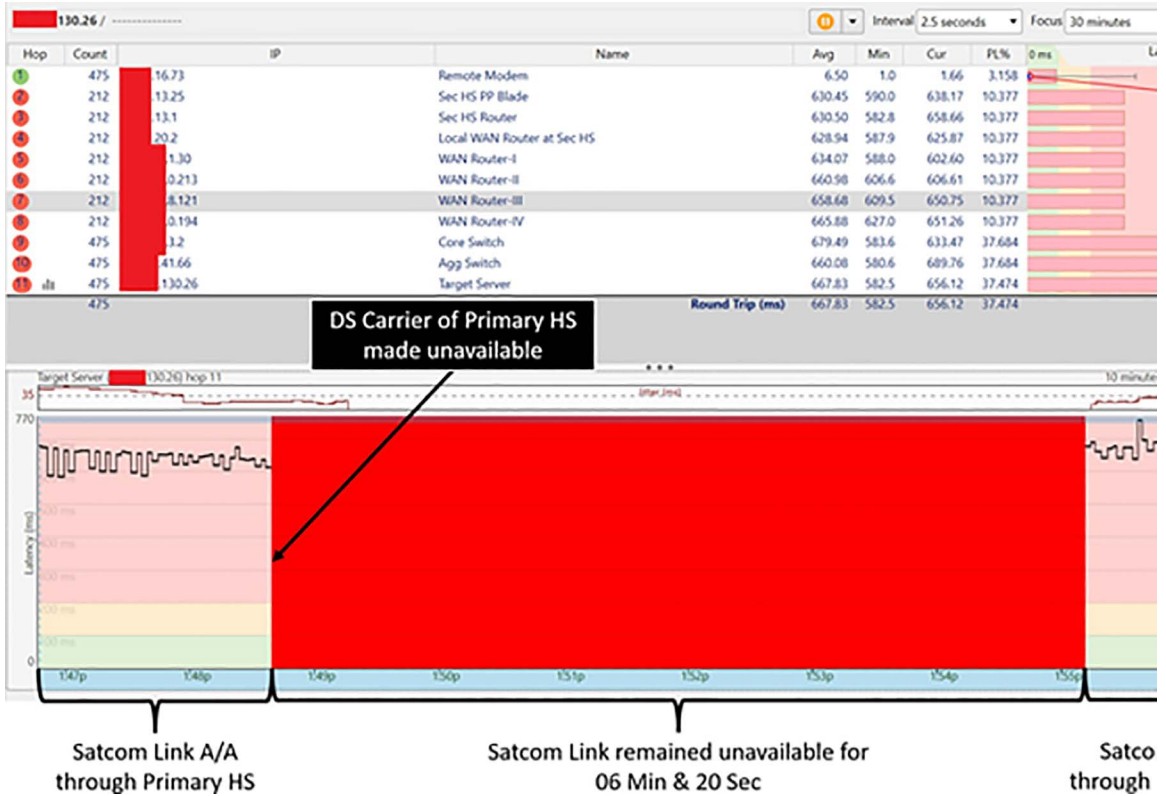

**Fig 5. Switching time from parent HS to secondary HS.**

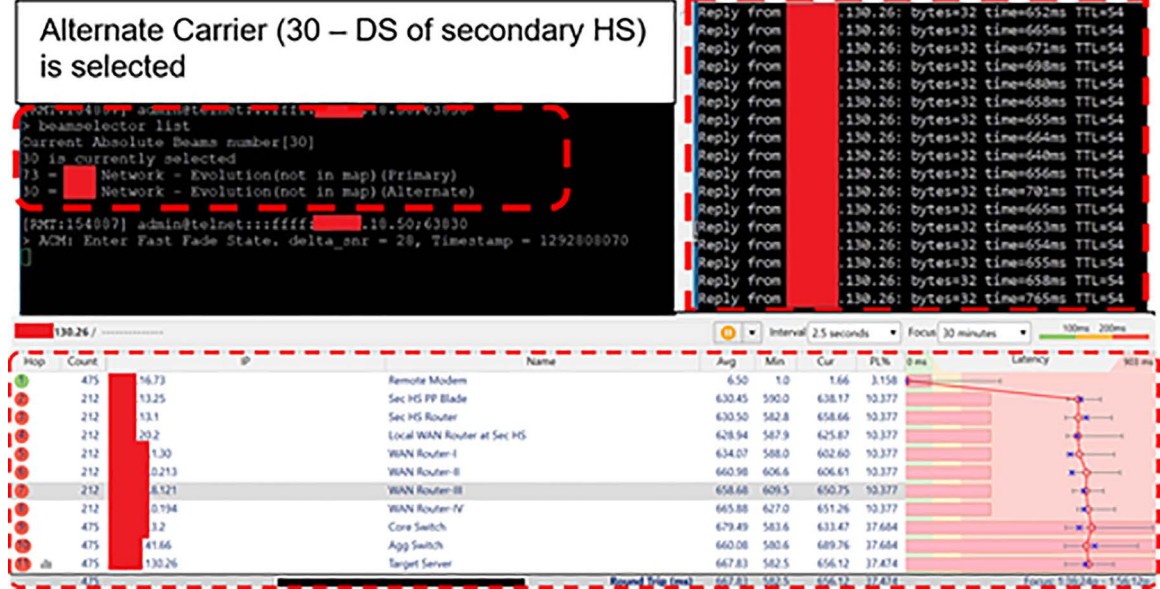

**Fig 6. Remote modem IP traffic routed through secondary HS.**

**Table 3. Traffic route while satcom link is through secondary HS.**

| No of Hop | NW Element | NE IP Address |
|---|---|---|
| 1 | Remote Modem | xx.xx.16.73 |
| 2 | Secondary HS PP Blade | xx.xx.13.25 |
| 3 | Secondary HS Router | xx.xx.13.1 |
| 4 | Local WAN Router | xx.xx.20.2 |
| 5 | WAN Router-I | xx.xx.1.30 |
| 6 | WAN Router-II | xx.xx.0.213 |
| 7 | WAN Router-III | xx.xx.8.121 |
| 8 | WAN Router-IV | xx.xx.0.194 |
| 9 | Core Switch | xx.xx.3.2 |
| 10 | Aggregation Switch | xx.xx.41.66 |
| 11 | Target Server | xx.xx.130.26 |

Remote Modem and the IP traffic route from user terminal to the Target Server while Satcom link was established through the secondary HS.

## Test/trials with minimum timeout setting

In the second phase of trials, timeout setting for searching primary DS frequency carrier before shifting to secondary DS carrier was changed from default 05 Minutes (300 Seconds) to 02 Minutes (120 Seconds). During testing with these settings, it was observed that once primary DS carrier became unavailable, the Remote Modem took 03 Minute & 20 Seconds to establish Satcom Link with the secondary HS. 02 Minutes for searching Primary Carrier before shifting to the Alternate Carrier were taken as per changed settings. In addition, acquisition of DS carrier of secondary HS, registration with the secondary HS and establishing communication link took 01 Minute and 20 Sec. Thus, time taken for complete shifting cycle from parent HS to a secondary HS was reduced from 06 Minutes & 20 Sec to 03 Minute & 20 Seconds. Fig 7 presents results of test/ trials and depicts the time taken by the Remote Modem to shift from primary to secondary HS under newly defined timeout value.

## Discussion

Key dividends/ results accrued by utilizing Unique Option File to implement the proposed automatic shifting mechanism are discussed in following paragraphs: -

### Automatic shifting

During the tests/ trials, automatic shifting of a Satcom Link from parent HS to a secondary HS was achieved without utilizing iDirect Global NMS. The automated switching of the Satcom Link will help in implementation of redundancy/ DR Mode in an iDirect Satcom Network; thus, ensuring reliability of the satcom-based communication links in case of a contingency/ disaster situation at the parent HS and vice-versa.

### Reduced switching time

During the trials/ testing, it was observed that the Remote Modem took 01 Minutes & 20 Seconds to establish Satcom Link with the secondary HS. In addition, 02 Minutes for searching Primary Carrier before shifting to the Alternate Carrier were taken with minimum timeout settings. Thus, time taken for complete shifting cycle for a Remote Modem from parent HS to a secondary HS, was 03 Minutes & 20 Seconds [02 Min (for Carrier Search) + 01 Min & 20

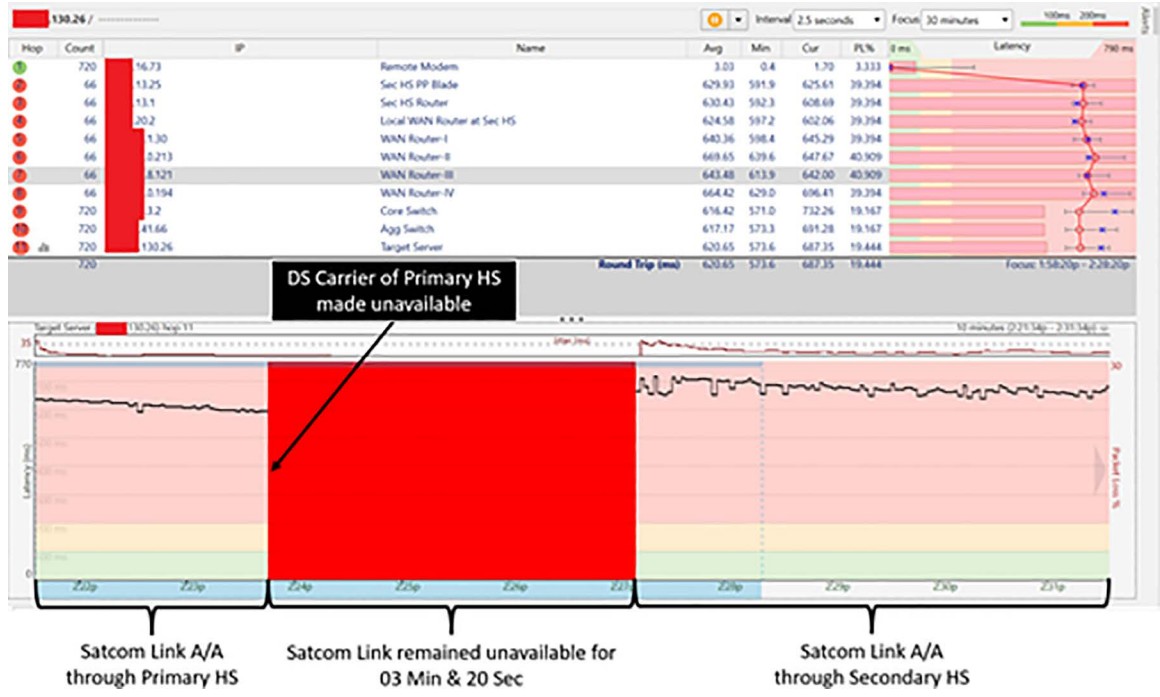

**Fig 7. Switching time from parent HS to secondary HS with 02 minutes timeout.**

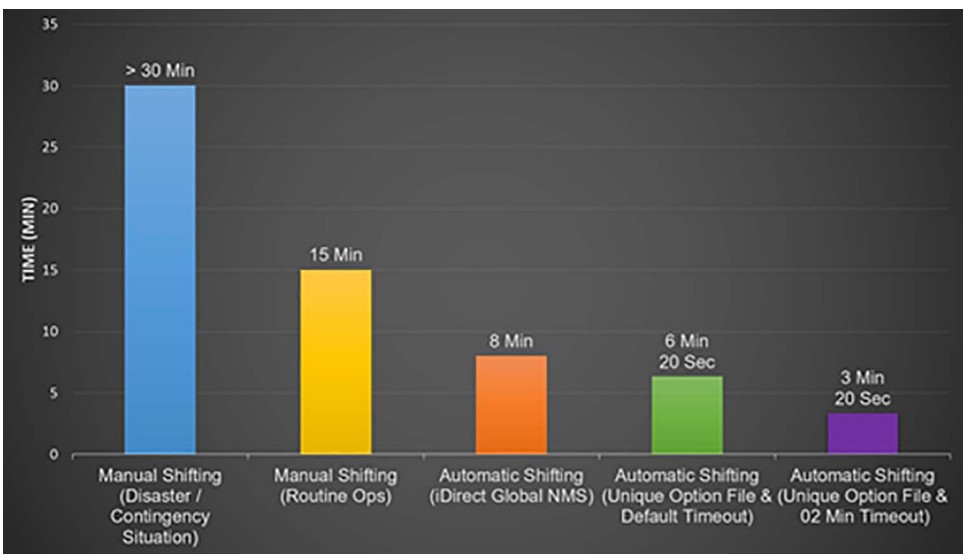

**Fig 8. Time taken for establishment of satcom link with different shifting methods.**

Sec (for Network Acquisition/ Registration)] which is significantly lesser than the time taken during manual shifting of Satcom links; thus, resulting in an efficient shifting of satcom-based communication links from one HS to another HS. Fig 8 presents a comparison of time taken while utilizing different methods for shifting of a Satcom Link from one HS to the other HS.

## Cost saving

The above-mentioned results on automatic shifting of Satcom Links were achieved on a real iDirect Satcom Network without deploying iDirect Global NMS. Therefore, procurement of iDirect Global NMS will not be required to implement DR Mode for achieving automatic switching of Satcom Links; thus, significant amount of capital investment will be saved.

A comparison of the available methodologies with the proposed solution is presented in Table 4. From the data, presented in the table, it is clearly visible that switching methodology employed using a Unique Option File has produced significantly improved results. Firstly, the method provides automatic shifting of a Satcom link from the primary HS to a secondary HS in an iDirect Satcom Network. Secondly, the Satcom Modem took much lesser time to establish Satcom link through the standby Hub Station as compared to the time taken with all the conventional techniques. In addition, all these results have been produced without utilizing the iDirect Global NMS; thus, eliminating the need for any additional hardware deployment. The implementation enables the Satcom Remote Terminals to automatically detect failures and initiate switchover procedures with no human intervention, resulting in an efficient shifting of Satcom-based communication links from one HS to another.

## Implications and applications of proposed algorithm

The optimization technique proposed in the research paper has significant implications for various satellite communication sectors. Its ability to provide rapid, automatic redundancy and failover through the use of the Unique Option File ensures reliable satellite links, even in the event of Hub Station (HS) failures. This section explores its potential applications across multiple satellite communication sectors:

## Telecommunication networks

The global demand for reliable mobile communication, especially in remote and underserved areas, is increasing. Many mobile communication providers rely on satellite backhaul to connect remote cellular base stations to the core network. In this context, the proposed optimization technique offers several advantages.

- **Improving Reliability for Mobile Backhaul:** Mobile networks, particularly in rural and hard-to-reach areas, often depend on satellite backhaul to transmit data from cellular towers to the core network. The proposed redundancy mechanism ensures that, in case of a failure at the primary HS, the satellite modem automatically shifts to a secondary HS, reducing downtime. This automatic switching eliminates the need for manual intervention, which can be delayed due to the remote nature of many sites.

**Table 4. Comparison of the available shifting methodologies with the proposed unique option file solution.**

| S No | Description | Type | Time Req for Shifting | Other Additional Requirements |
|---|---|---|---|---|
| 1. | Shifting during Routine Ops | Manual | 15 Min | HS NMS & Operator |
| 2. | Shifting during Contingency | Manual | 30 Min | Management Terminal along with necessary files, SWs & Operator at Remote Site |
| 3. | Shifting with Global NMS | Automatic | 8 Min | iDirect Global NMS, additional HW & changes in NW |
| 4. | Shifting with Proposed Unique Option File | Automatic | 3 Min 20 Sec | Nil |

- **Consistent Service for Remote Users:**By optimizing the time it takes for the modem to switch between Hub Stations (from 6 minutes to 3 minutes 20 seconds), the technique ensures minimal service disruption for users who rely on satellite connectivity as part of the broader telecommunications network. This is particularly important for mobile operators providing services in rural areas where fiber or microwave backhaul is unavailable.

- **Scalability for Expanding Networks:**As mobile operators expand into more remote regions, the need for scalable, reliable satellite backhaul becomes more critical. The proposed optimization technique can easily scale with expanding networks, as it does not require additional hardware or costly network upgrades, making it a cost-effective solution for telecom providers.

This technique ensures that mobile communication providers can maintain reliable service for customers even during Hub Station outages, improving overall network resilience and customer satisfaction.

## Emergency services

In disaster recovery and humanitarian relief efforts, uninterrupted communication is critical for coordinating rescue operations, sharing real-time information, and delivering aid. Satellite communication often serves as the primary or backup communication method in these scenarios due to its wide coverage and reliability. The proposed optimization technique can greatly enhance the effectiveness of emergency services through:

- **Rapid Redundancy in Disaster Situations:**Natural disasters such as earthquakes, floods, or hurricanes often damage terrestrial communication infrastructure, leaving emergency responders reliant on satellite networks. The technique's rapid automatic failover mechanism ensures that, in the event of a Hub Station failure, communication links are quickly re-established with a secondary HS. This failover happens in under 4 minutes, minimizing the communication downtime that can be detrimental to disaster response efforts.

- **Maintaining Connectivity in Remote Locations:**Humanitarian operations frequently take place in remote regions where terrestrial infrastructure is non-existent or unreliable. The redundancy mechanism ensures that satellite-based communication links remain operational, providing a lifeline for coordination efforts. This is particularly valuable in scenarios where real-time communication is needed to direct resources and manage logistics.

- **Efficient Network Recovery:**The ability of the system to switch automatically without manual intervention is critical when skilled personnel are not immediately available on-site to handle network configuration. The optimized option file ensures that remote terminals can continue to function autonomously, even when the primary Hub Station is compromised.

## Military communications

Military operations require secure, reliable, and low-latency communication for command, control, and coordination. Satellite communication plays an essential role in military networks, especially for troops deployed in remote or hostile environments where traditional terrestrial infrastructure may not be viable. The proposed optimization technique offers several critical advantages for military applications:

- **Low-Latency, High-Reliability Communication:**Military operations depend on real-time data exchange for mission-critical operations, including command and control, surveillance,

and reconnaissance. The optimization technique ensures that communication links between military satellites and ground stations remain intact even during Hub Station failures. The reduced switching time (to 3 minutes 20 seconds) minimizes the risk of communication breakdowns that could jeopardize military operations.

- **Enhancing Tactical Mobility:**Military forces are often mobile, requiring satellite systems that can dynamically adjust to changing locations and operational demands. The ability to automatically switch between Hub Stations ensures that satellite links remain reliable, even as troops move between different theaters of operation. This capability is particularly valuable for ensuring continuous communication during rapid deployments or when operating in challenging environments.

- **Operational Security and Redundancy:**The proposed technique provides additional security by ensuring communication continuity during HS failures, which could be caused by cyber-attacks, jamming, or other hostile activities. The automatic failover mechanism ensures that secure communication links are quickly restored, preventing potential exploitation by adversaries.

- **Support for Remote Command Posts:**Military command posts located in remote areas often rely on satellite communication to maintain contact with central command. The proposed technique ensures these command posts have access to a stable, high-reliability satellite link, which is critical for real-time decision-making and operational coordination.

## Other potential applications

The optimization technique also has broader applications across other sectors:

- **Maritime Communication:**Ships and offshore platforms rely on satellite communication for navigation, weather updates, and operational coordination. This technique ensures consistent satellite links, even in adverse weather or in case of HS failures, which is critical for maritime safety and operational efficiency.

- **Aviation:**Aircraft increasingly rely on satellite communication for in-flight connectivity and real-time updates. The proposed optimization can ensure seamless communication during flights, especially in polar or oceanic regions where terrestrial links are unavailable.

- **Broadcasting:**Satellite networks are widely used in broadcasting, particularly for live events. The rapid redundancy mechanism ensures continuous broadcasting signals even if one Hub Station fails, maintaining service quality and reducing downtime.

In summary, the proposed optimization technique holds significant potential across various satellite communication sectors, from improving mobile backhaul reliability to enhancing military communication capabilities. By offering rapid redundancy, automatic failover, and minimal disruption during Hub Station failures, the technique provides a robust, scalable solution for industries where uninterrupted satellite connectivity is crucial. Its wide-ranging applicability makes it a valuable advancement in ensuring reliable satellite communications in both routine and critical operations.

## Limitations of proposed technique

While the proposed optimization technique offers significant improvements in redundancy and automatic failover for satellite communication systems, there are several limitations that need to be addressed to provide a more comprehensive view of its applicability and performance. These limitations include the reliance on specific hardware configurations, potential

latency in switching time, and the challenges posed by certain satellite architectures such as Low Earth Orbit (LEO) systems.

## Reliance on specific hardware configurations

The current approach heavily relies on the use of iDirect modems and iDirect Hub Stations for satellite communication. While iDirect is a prominent provider of satellite communication equipment, this specificity limits the generalizability of the solution to other systems. Satellite networks that use different modems or platforms may not be able to adopt this technique without significant modifications. For instance:

- **Hardware Dependency:** The use of iDirect-specific features, such as the Unique Option File and Derived Identification (DID), makes the technique dependent on the iDirect ecosystem. While this ensures seamless integration within iDirect systems, other satellite communication platforms (such as those using Newtec, Hughes, or Viasat hardware) would require substantial adjustments to implement similar functionality.

- **Limited Vendor Compatibility:** Networks using hardware from multiple vendors or custom-built systems may face integration challenges. The method would require adaptation to fit into multi-vendor environments, which might entail the development of new protocols and algorithms to enable cross-platform compatibility.

While highly effective in iDirect-based networks, the approach may not be suitable for 665 satellite communication systems using different hardware configurations. To increase 666 the broader applicability of this method, future research and development should focus 667 on adapting the solution to work with a wider range of modem and Hub Station technologies.

## Switching time and latency

Although the switching time in the proposed solution has been optimized (from over 6 minutes to 3 minutes and 20 seconds), this latency may still be too high for certain real-time applications. Some use cases where continuous, uninterrupted communication is critical may experience noticeable disruptions during the switchover process:

- **Real-time Applications:** Applications such as video streaming, online gaming, or voice over IP (VoIP) require minimal latency to ensure a smooth user experience. While the reduced switchover time represents a significant improvement, a gap of 3 minutes can still cause interruptions that could degrade the quality of real-time services, particularly in scenarios where continuous communication is essential.

- **Critical Communications:** In some environments, such as financial services or high-frequency trading, even a few seconds of downtime could result in financial loss or missed opportunities. In these cases, the current switchover time may not be acceptable, and further optimization or faster switching protocols would be needed to support such mission-critical applications.

The method greatly improves on existing manual or slower switching mechanisms, but the remaining latency in the switchover process limits its effectiveness for real-time, latency-sensitive applications. Future work could focus on reducing the failover time even further, possibly by preemptively identifying failures or using faster frequency acquisition techniques.

## Applicability to LEO and non-geostationary satellite systems

The method's focus on GEO (Geostationary Earth Orbit) satellites poses another limitation. LEO (Low Earth Orbit) and MEO (Medium Earth Orbit) satellites operate in more dynamic

environments, requiring more frequent handovers between satellites and ground stations. The current solution may face challenges in adapting to the following scenarios:

- **Frequent Handover in LEO Systems:** LEO satellites move rapidly in their orbits relative to the Earth's surface, requiring ground stations to switch frequently between satellites to maintain connectivity. This dynamic nature presents a challenge for the proposed method, as it relies on a relatively static relationship between the satellite and ground station. The current optimization technique, designed for failover between stationary GEO satellites and ground stations, may not account for the frequent handovers that are inherent to LEO networks.

- **Orbit Variability:** LEO and MEO satellites have lower altitudes and shorter lifespans compared to GEO satellites. These systems may require more agile and responsive failover mechanisms, as satellites may exit coverage areas quickly. The proposed technique may need further investigation to determine whether its approach to switching between Hub Stations can handle the rapid, often unpredictable changes in satellite availability associated with non-geostationary orbits.

The technique is well-suited for GEO satellite systems, where the satellite remains stationary relative to the ground station. However, its applicability to LEO and MEO systems, where frequent handovers are required, is limited without further adaptations.

Future research should focus on enhancing the failover mechanism for use in highly dynamic satellite constellations such as LEO orbits.

## Scalability and network complexity

As satellite communication networks become more complex, particularly with the rise of mega-constellations (such as Starlink and OneWeb), the proposed technique may face scalability challenges. While the approach works well for smaller or moderately sized networks, more investigation is needed to assess how the solution performs in larger networks with hundreds or thousands of satellites and ground stations:

- **Mega-constellation Networks:** These networks introduce additional complexity due to the large number of satellites in operation, each requiring frequent handovers and failover mechanisms. The scalability of the proposed solution in such scenarios may be limited without further enhancements to handle the increased load of network switches, frequency changes, and satellite-to-ground station transitions.

While the technique provides redundancy for small to medium satellite networks, its scalability in large-scale constellations may be limited. Additional development and testing are required to ensure its effectiveness in handling the complexity of mega-constellations.

## Future work

The proposed optimization technique presents a significant advancement in satellite communication redundancy, there are several avenues for further study to expand its applicability and efficiency across different satellite architectures and emerging technologies. Moreover in order to overcome some of the limitations presented in the proposed technique future research could focus on the following areas

- **FirmWare Development:** One of the future research directions is to develop a Firmware to mitigate the challenges of software mismatches. The developed firmware should allow for multi-version support, enabling the satellite modems to recognize different versions and adjust accordingly during the handover process.

## Performance impact and network congestion

During the automatic switching process, potential network congestion, particularly in cases where multiple remote terminals initiate switching simultaneously will occur. To resolve this situation following measures may be explored in future research

- **Testing Multiple Switching Events:**In this paper testing of the proposed technique is focused on single terminal switching, however additional tests that simulate multiple remote terminals switching simultaneously are planned for future updates. These results will better understand the effects on bandwidth, latency, and potential bottlenecks in a more congested network environment.

- **Resource Allocation and Traffic Management:**The future work will include incorporating traffic management techniques, such as bandwidth reservation or prioritization mechanisms, to ensure critical communication flows remain uninterrupted during high-traffic scenarios.

## Latency and scalability considerations

When multiple terminals switch to the secondary hub simultaneously, there could be a latency spike due to the load on both the Network Management System (NMS) and the hub station. To overcome this challenge following process can be adapted to the proposed technique.

- **Optimized Switching Timers:**The proposed technique experimented with adjusting the timeout settings for carrier searching, which reduced the overall switching time from over 6 minutes to 3 minutes. However, this process can be further optimized in the future by introducing adaptive timers based on the network load, ensuring that terminals stagger their switchovers to prevent bottlenecks.

- **Parallel Switching Capability:**In the future the system will be tested to ensure that it can handle multiple switchovers in parallel without significantly impacting overall network performance. Implementing more robust load-balancing mechanisms in the NMS and distributing the switching load between different hub stations will be essential in preventing latency spikes.

## Scalability in diverse SATCOM networks

The proposed technique is tested in a controlled network environment. We plan to extend testing to more complex SATCOM network environments that involve varying traffic loads, different types of remote terminals, and multiple simultaneous handover events. This will provide more data on the system's robustness and scalability.

## Machine learning models for predictive failure detection

One of the challenges of satellite communication is identifying potential failures before they occur. Integrating machine learning (ML) models into the system could significantly enhance its ability to predict failures in satellite ground segments and preemptively switch to backup Hub Stations.

- **Predictive Analytics for Failures:**By using historical data on Hub Station performance, satellite health, and network traffic patterns, ML algorithms could be trained to identify early warning signs of potential failures. These signs might include decreasing signal strength, network congestion, or minor hardware malfunctions that often precede complete failures.

- **Preemptive Switching:** With predictive failure detection, the system could initiate the failover process before a Hub Station failure fully disrupts communication. This would not only reduce the switching time but also prevent potential service interruptions for real-time and critical applications.

- **Self-Learning Systems:** Over time, the system could refine its predictive capabilities through continuous learning, allowing the model to become more accurate in anticipating failures based on various operational factors. This would make the redundancy mechanism more proactive, rather than reactive, ensuring even higher reliability.

## Integration with satellite-based internet of things (IoT) systems

The rapid growth of the Internet of Things (IoT) is driving the need for satellite-based communication solutions that can support a massive number of connected devices. IoT systems often rely on satellites for connectivity in remote areas, where terrestrial networks are unavailable. Future research should focus on how the proposed redundancy technique can be optimized for IoT applications, particularly in terms of bandwidth management and device reliability.

- **Optimized Bandwidth Allocation:** IoT systems typically involve many devices transmitting small amounts of data, but the sheer volume of devices can create congestion in the satellite network. The redundancy mechanism could be adapted to ensure that the switchover process does not result in bandwidth bottlenecks and that connected devices continue to operate seamlessly even during failovers.

- **Reliability for Critical IoT Applications:** In sectors such as agriculture, environmental monitoring, and smart cities, IoT devices rely heavily on uninterrupted satellite connectivity. The failover mechanism could be optimized to prioritize critical IoT devices during failover events, ensuring that essential systems remain operational even during network disruptions.

- **Disaster Recovery and Resilience:** In emergency scenarios, such as natural disasters or infrastructure failures, maintaining an uninterrupted communication link is essential for coordinating rescue efforts and disaster management operations. Our results demonstrate that the failover time using the proposed solution is reduced to 3 minutes and 20 seconds, compared to over 6 minutes with conventional methods. This reduction in downtime ensures faster re-establishment of communication links, which is crucial for time-sensitive operations. Unlike existing systems that require manual intervention, our solution allows fully automated failover, ensuring that no human error or delays affect the recovery process. This approach could be instrumental in sectors like emergency response, where satellite communication is often the only means of coordination in remote or disaster-stricken areas.

- **Cost-Effectiveness in Commercial Applications:** Many Satcom providers, particularly those operating in developing regions or remote areas, face significant budget constraints when it comes to deploying redundant systems. Traditional methods, such as the implementation of iDirect Global NMS, can involve high capital expenditure due to additional hardware requirements. Our results indicate that the proposed solution eliminates the need for iDirect Global NMS while still providing automatic redundancy. This not only cuts down on upfront costs but also reduces the operational complexity of managing a redundant network. The practical impact is that smaller satellite operators and those with limited budgets can implement a cost-effective solution without sacrificing performance or reliability.

- **Scalability for Massive IoT Deployments:** As IoT networks grow in scale, future research could focus on how to enhance the failover mechanism to handle the demands of millions of IoT devices simultaneously. This might involve developing more sophisticated prioritization algorithms that manage device traffic during Hub Station switching and ensuring that latency-sensitive applications, such as real-time monitoring, receive prioritized bandwidth.

## Conclusion

Reliability of satellite-based communication links can be affected by various factors, including the unavailability of a Satcom Hub Station. This fact is also relevant for an iDirect Satcom Network, where remote terminals rely on a Hub Station for communication. In the event of a Hub Station malfunction, its remote terminals become standalone, losing Satcom link until they can shift to another Hub Station. Various manual and automatic methods are available for shifting remote terminals, but they require significant time or hardware/software resources to implement. Therefore, an optimized shifting mechanism was needed to enhance the reliability of Satcom links through automatic switching without major hardware or software changes to the network.

From the research and results presented-above, it can be concluded that the automatic switching methodology employed using a Unique Option File produced significantly improved results. The efficiency of proposed technique out performs traditional switching techniques in many ways. Automatic shifting of a Satcom link from the primary HS to a secondary HS was achieved in an iDirect Satcom Network without utilizing the iDirect Global NMS. The implementation enables the Satcom remote terminals to automatically detect failures and initiate switchover procedures with no human intervention. Additionally, the Satcom modem took significantly reduced time to establish Satcom link through the standby Hub Station as compared to conventional techniques. This resulted in efficient shifting of Satcom-based communication links from one HS to another.

Implementing this concept in an iDirect Satcom Network will significantly enhance the reliability of Satcom links by enabling efficient shifting of the Satcom links from primary Hub Station to a secondary Hub Station in case of a contingency or disaster situation at the primary Hub Station.

## Author contributions

**Conceptualization:** Bilal Muhammad Khan.

**Data curation:** Bilal Muhammad Khan, Muhammad Raheel.

**Formal analysis:** Bilal Muhammad Khan.

**Investigation:** Bilal Muhammad Khan, Muhammad Raheel.

**Methodology:** Muhammad Raheel.

**Resources:** Bilal Muhammad Khan.

**Software:** Bilal Muhammad Khan.

**Supervision:** Bilal Muhammad Khan.

**Validation:** Bilal Muhammad Khan.

**Visualization:** Bilal Muhammad Khan.

**Writing – original draft:** Muhammad Raheel.

**Writing – review & editing:** Bilal Muhammad Khan.

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
