## [Decision Letter · Decision Letter 0]

3 Sep 2024

PONE-D-24-33931Optimization of Satellite-based Communication LinksPLOS ONE

Dear Dr. Khan,

Thank you for submitting your manuscript to PLOS ONE. After careful consideration, we feel that it has merit but does not fully meet PLOS ONE’s publication criteria as it currently stands. Therefore, we invite you to submit a revised version of the manuscript that addresses the points raised during the review process.

We look forward to receiving your revised manuscript.

Kind regards,

Muhammad Zulkifl Hasan, PhD

Academic Editor

PLOS ONE

[NA]. 

Additional Editor Comments:

The manuscript titled "Optimization of Satellite-based Communication Links" presents a comprehensive approach to enhancing the reliability and efficiency of satellite communication networks, particularly focusing on the automatic shifting of remote terminals between Hub Stations to ensure continuous service. The authors propose an innovative method using a Unique Option File that enables seamless switching without significant hardware or software changes. The methodology is validated through practical tests, demonstrating substantial improvements in switching time and cost-effectiveness compared to existing methods.

To improve the manuscript, consider providing more detailed explanations of the underlying principles of the proposed algorithm and how it integrates with existing satellite communication technologies. Additionally, expanding the discussion on the broader implications and potential applications of this optimization technique across different satellite networks would strengthen the manuscript's contribution to the field. Clarifying the limitations and future research directions would also provide a more comprehensive perspective for readers.

Reviewers' comments:

Reviewer's Responses to Questions

**Comments to the Author**

1. Is the manuscript technically sound, and do the data support the conclusions?

Reviewer #1: Yes

Reviewer #2: Partly

2. Has the statistical analysis been performed appropriately and rigorously? 

Reviewer #1: Yes

Reviewer #2: No

3. Have the authors made all data underlying the findings in their manuscript fully available?

Reviewer #1: Yes

Reviewer #2: No

4. Is the manuscript presented in an intelligible fashion and written in standard English?

Reviewer #1: Yes

Reviewer #2: No

5. Review Comments to the Author

Reviewer #1: The manuscript titled "Optimization of Satellite-based Communication Links" demonstrates a significant effort to address the challenges of redundancy and disaster recovery in satellite communication (SATCOM) networks. While the research presents an innovative solution using a unique option file for automatic shifting between primary and secondary hub stations without requiring significant hardware changes or the iDirect Global NMS, there are several weaknesses in the approach that need to be addressed.

Firstly, the methodology heavily relies on the assumption that both the primary and secondary hub stations operate on the same iDirect Evolution Software version. This creates a potential limitation as any software updates or version mismatches could render the proposed solution ineffective. Additionally, the study lacks a detailed analysis of the performance impacts or potential network congestion that may arise during the automatic switching process. The research also does not adequately consider scenarios where multiple remote terminals simultaneously initiate the switching process, which could lead to network bottlenecks or increased latency. These limitations suggest that while the proposed solution is promising, further testing and optimization are required to ensure its robustness and scalability in diverse SATCOM network environments.

Reviewer #2: Clarity and Structure of the Introduction:

The introduction section is quite dense, with many technical details presented early on. Consider breaking down the introduction into more digestible segments, where the broader context and significance of the study can be established before delving into the specific technical challenges and solutions. This will make it easier for readers unfamiliar with the subject to understand the importance of the research.

Literature Review Integration:

While the manuscript references a wide array of studies, the integration of these references could be more cohesive. Currently, the references feel somewhat disjointed. A more synthesized review of the literature, focusing on how the proposed work builds upon or diverges from existing studies, would strengthen the manuscript's foundation.

Detailed Methodological Justification:

The methodology section provides a general overview of the processes used. However, it would benefit from a more detailed explanation of why certain methods were chosen over others. For instance, the decision to not use iDirect Global NMS could be further elaborated with a comparison of potential alternatives and their limitations. This will help to justify the proposed approach more convincingly.

Validation and Generalization of Results:

The manuscript presents a single case study as validation of the proposed model. While the results are promising, the manuscript would be stronger if it included multiple test cases or scenarios to demonstrate the robustness and generalizability of the model across different conditions. This could include varying environmental factors, network configurations, or hardware setups.

Quantitative Analysis of Performance Improvements:

The results section discusses the time savings achieved through the proposed model. However, a more in-depth quantitative analysis, possibly with statistical validation, could provide stronger evidence of the model’s superiority. For example, comparing the performance metrics across several trials and providing a detailed analysis of the variance would add rigor to the findings.

Technical Details on Implementation:

The manuscript mentions the development of a Unique Option File for registration of Satcom Modems. More detailed information on the implementation of this file, such as the specific parameters configured and how they interact with the network, would be valuable. Additionally, a flowchart or diagram illustrating the process could enhance reader comprehension.

Discussion of Limitations and Future Work:

While the manuscript briefly mentions the need for further development, it lacks a detailed discussion of the current limitations of the proposed model. Identifying areas where the model might struggle or scenarios it may not cover would provide a more balanced perspective. Additionally, outlining concrete steps for future research would guide subsequent studies and demonstrate the authors’ awareness of the field’s evolving challenges.

Figures and Tables:

The figures and tables included in the manuscript are useful, but their captions could be more descriptive. Each figure and table should be able to stand alone, with the captions providing enough context for readers to understand the content without referring back to the main text.

Language and Technical Jargon:

The manuscript is written in a highly technical style, which may limit its accessibility to a broader audience. Simplifying some of the technical jargon and providing brief explanations of complex terms when first introduced could make the manuscript more inclusive to readers from different backgrounds.

Ethical and Practical Considerations:

The manuscript does not address any ethical or practical considerations related to the deployment of the proposed system in real-world environments. Discussing potential risks, such as data security concerns or operational challenges, would provide a more comprehensive view of the implications of this research.

By addressing these areas, the manuscript could significantly improve in clarity, rigor, and overall impact. The research presented is valuable, but refining these aspects will enhance its contribution to the field of satellite communication optimization.

6. PLOS authors have the option to publish the peer review history of their article (what does this mean? ). If published, this will include your full peer review and any attached files.

**Do you want your identity to be public for this peer review?** For information about this choice, including consent withdrawal, please see our Privacy Policy .

Reviewer #1: **Yes: ** Muhammad Zunnurain Hussain

Reviewer #2: No

---

## [Author Response · Author response to Decision Letter 1]

25 Sep 2024

Editor’s and Reviewer’s Comments

The Authors would like to take this opportunity to appreciate Editor and Reviewers Constructive feedback and made sure that all the comments made by the worthy persons are addressed to the best of Author’s knowledge. Please find below Comments by Comments Responses. These are also being incorporated in the revised manuscript uploaded by the Authors.

Editor/Reviewers Comments Response

Editor’s Comments 1. To improve the manuscript, consider providing more detailed explanations of the underlying principles of the proposed algorithm A complete new section is added in the revised manuscript under the heading of “Proposed Algorithm” on Page 06, Line # 205-289.

Moreover a flow chart of the proposed algorithm is also added as Figure #01 on Page 08, Line#277 of the revised manuscript.

2. Explain how the proposed technique integrates with existing satellite communication technologies. A complete new section is added in the revised manuscript under the heading of “Integrating proposed technique with Existing Satellite Technologies” on Page 09, Line # 312-406.

3. Additionally, expanding the discussion on the broader implications and potential applications of this optimization technique across different satellite networks would strengthen the manuscript's contribution to the field. A complete new section is added in the revised manuscript under the heading of “Implications and Applications of Proposed Algorithm” on Page 17, Line # 533-640.

4. Clarifying the limitations and future research directions would also provide a more comprehensive perspective for readers.

A complete new section is added in the revised manuscript under the heading of “Limitations of Proposed Technique” on Page 20, Line # 641-731.

A complete new section is added in the revised manuscript under the heading of “Future Work” on Page 22, Line # 732-817.

Reviewer # 1 Comments 1. The methodology heavily relies on the assumption that both the primary and secondary hub stations operate on the same iDirect Evolution Software version. This creates a potential limitation as any software updates or version mismatches could render the proposed solution ineffective. The Authors appreciate the reviewer highlighting the assumption related to both the primary and secondary hub stations operating on the same iDirect Evolution Software version. We acknowledge this as a potential limitation, however, since the algorithm is developed on running hardware actual system this limitation can be justified and further rectified as discussed on Page#09, Line # 291-311; Under the headings of “Software Version Compatibility & Mitigation of Version Mismatch Risks” of revised manuscript.

Furthermore a subsection of “Frame ware Development” under the section of “Future Work” on Page #22, Line#733-741, elaborates in detail how to eliminate such limitations in future extension of the proposed work.

2. The study lacks a detailed analysis of the performance impacts or potential network congestion that may arise during the automatic switching process. The research also does not adequately consider scenarios where multiple remote terminals simultaneously initiate the switching process, which could lead to network bottlenecks or increased latency. These limitations suggest that while the proposed solution is promising, further testing and optimization are required to ensure its robustness and scalability in diverse SATCOM network environments.

Authors appreciate reviewer’s insights into the potential performance impacts and network congestion during the automatic switching process. Also acknowledge the fact that further testing is essential to analyze and ensure the scalability and robustness of our proposed solution. All these concerns are addressed under a complete new section added in the revised manuscript “Future Work” on PAGE #22, Line #732-775.

Reviewer # 2 Comments

The methodology section provides a general overview of the processes used. However, it would benefit from a more detailed explanation of why certain methods were chosen over others. For instance, the decision to not use iDirect Global NMS could be further elaborated with a comparison of potential alternatives and their limitations. This will help to justify the proposed approach more convincingly. Details along with explanation of why certain methods were chosen over others already mentioned under the heading “Hub Station Redundancy in an iDirect Network”. Comparison of the available methodologies is included in “Table 1”, Page#4, and Line #117-118 in the revised manuscript.

Moreover since iDirect Satcom equipment is based on proprietary technologies and algorithms unique to the equipment manufacturer i.e. ST Engineering. The only solution provided by the vendor for implementing automatic shifting of remotes between two Hub Station is the implementation of iDirect Global NMS. Since, no alternatives are available; therefore, comparison has only been made with the possible Manual Switching mechanisms / alternatives.

The manuscript presents a single case study as validation of the proposed model. While the results are promising, the manuscript would be stronger if it included multiple test cases or scenarios to demonstrate the robustness and generalizability of the model across different conditions. This could include varying environmental factors, network configurations, or hardware setups. Validation of the proposed model has been carried out on a real iDirect Satcom Network. Since, the facility used for the testing is an operational iDirect Network and testing required disruption of services by Switching ‘Off’ Downstream Frequency Carrier of the Primary Hub Station; therefore, multiple test cases or scenarios could not be implemented as same would have resulted in unavailability of the Satcom Network for users.

However, extensive testing and validation of the proposed switching mechanism under various scenarios and load conditions is being suggested under the section of “Future Work” on Page#22, Line 732-815. Ensuring that the proposed automatic switching mechanism performs reliably under all circumstances.

The results section discusses the time savings achieved through the proposed model. However, a more in-depth quantitative analysis, possibly with statistical validation, could provide stronger evidence of the model’s superiority. For example, comparing the performance metrics across several trials and providing a detailed analysis of the variance would add rigor to the findings. Validation of the proposed model has been carried out on a real iDirect Satcom Network. Since, the facility used for the testing is an operational iDirect Network and testing required disruption of services by Switching ‘Off’ Downstream Frequency Carrier of the Primary Hub Station; therefore, performance across several trials could not be tested as same would have resulted in unavailability of the Satcom Network for users. However, the methodology was tested for more than 20 times to validate the automatic switching during each test.

Extensive testing and validation of the proposed switching mechanism is being suggested under the section of “Future Work” on Page#22, Line 732-815.. This will ensure that the proposed automatic switching mechanism performs reliably under all circumstances.

The manuscript mentions the development of a Unique Option File for registration of Satcom Modems. More detailed information on the implementation of this file, such as the specific parameters configured and how they interact with the network, would be valuable. Additionally, a flowchart or diagram illustrating the process could enhance reader comprehension. As suggested by the reviewer key features of the developed solution are further enhanced and elaborated under the section “Optimized Shifting Model” on Page#04, Line # 130-204.

Furthermore a complete new section is added in the revised manuscript under the heading of “Proposed Algorithm” on Page 06, Line # 205-289.

Moreover a flow chart of the proposed algorithm is also added as Figure #01 on Page 08, Line#277 of the revised manuscript.

These sections further clarify and elaborate the working mechanism of the proposed technique.

While the manuscript briefly mentions the need for further development, it lacks a detailed discussion of the current limitations of the proposed model. Identifying areas where the model might struggle or scenarios it may not cover would provide a more balanced perspective. Additionally, outlining concrete steps for future research would guide subsequent studies and demonstrate the authors’ awareness of the field’s evolving challenges. As per reviewer suggestions:

A complete new section is added in the revised manuscript under the heading of “Limitations of Proposed Technique” on Page 20, Line # 641-731.

Complete new section added in the revised manuscript “Future Work” on PAGE #22, Line #732-775.

The manuscript is written in a highly technical style, which may limit its accessibility to a broader audience. Simplifying some of the technical jargon and providing brief explanations of complex terms when first introduced could make the manuscript more inclusive to readers from different backgrounds. The authors as per reviewer suggestions to the best of their knowledge where required explained Jargons and tried to make entire manuscript easily understandable by providing explanations to complex terms where ever possible in a balanced way.

The manuscript does not address any ethical or practical considerations related to the deployment of the proposed system in real-world environments. Discussing potential risks, such as data security concerns or operational challenges, would provide a more comprehensive view of the implications of this research. Implementation of the proposed solution, does not tailor / change iDirect's proprietary technology as well as communication algorithm; therefore, it does not result in a data security concern in addition to existing limitations of iDirect equipment.

---

## [Decision Letter · Decision Letter 1]

22 Oct 2024

PONE-D-24-33931R1Optimization of Satellite-based Communication LinksPLOS ONE

Dear Dr. Khan,

Thank you for submitting your manuscript to PLOS ONE. After careful consideration, we feel that it has merit but does not fully meet PLOS ONE’s publication criteria as it currently stands. Therefore, we invite you to submit a revised version of the manuscript that addresses the points raised during the review process.

We look forward to receiving your revised manuscript.

Kind regards,

Elochukwu Ukwandu, PhD

Academic Editor

PLOS ONE

Journal Requirements:

Additional Editor Comments:

**The Authors are to address comments from Reviewer #3 as shown below.**

The authors have put considerable effort into addressing the reviewers' comments. While the paper has undergone significant improvements, the authors are encouraged to make the following adjustments:

1. Despite the authors' revisions, certain sections still require more practical discussion and improved coherence in the results discussion.

2. The discussions regarding both technical and practical aspects, as well as comparisons to recent studies on this topic, need thorough examination.

3. For effective comparison of the proposed scheme with the other schemes, the variables of the presented results should start from almost the same steady state. However, this is not the case in this paper.

4. The references require updating and upgrading, as many of them are outdated. Additionally, some references need to be more practical, impactful, comprehensive, and robust.

https://doi.org/10.1016/j.optcom.2020.126219

https://doi.org/10.1049/ote2.12111

I recommend accepting the paper after minor revisions

Comments from PLOS Editorial Office: We note that one or more reviewers has recommended that you cite specific previously published works. As always, we recommend that you please review and evaluate the requested works to determine whether they are relevant and should be cited. It is not a requirement to cite these works. We appreciate your attention to this request.

Reviewers' comments:

Reviewer's Responses to Questions

**Comments to the Author**

1. If the authors have adequately addressed your comments raised in a previous round of review and you feel that this manuscript is now acceptable for publication, you may indicate that here to bypass the “Comments to the Author” section, enter your conflict of interest statement in the “Confidential to Editor” section, and submit your "Accept" recommendation.

Reviewer #1: All comments have been addressed

Reviewer #3: (No Response)

2. Is the manuscript technically sound, and do the data support the conclusions?

Reviewer #1: Yes

Reviewer #3: (No Response)

3. Has the statistical analysis been performed appropriately and rigorously? 

Reviewer #1: Yes

Reviewer #3: (No Response)

4. Have the authors made all data underlying the findings in their manuscript fully available?

Reviewer #1: Yes

Reviewer #3: (No Response)

5. Is the manuscript presented in an intelligible fashion and written in standard English?

Reviewer #1: Yes

Reviewer #3: (No Response)

6. Review Comments to the Author

Reviewer #1: The author has incorporated all the changes as desired. The revisions have been well addressed and significantly improved the clarity and overall quality of the manuscript. The methodology has been thoroughly explained, and the results are presented in a clear and concise manner. All previously noted concerns have been adequately resolved, and there are no outstanding issues related to dual publication, research ethics, or publication ethics. I am satisfied with the current version of the manuscript, and it is now ready for publication.

Reviewer #3: The authors have put considerable effort into addressing the reviewers' comments. While the paper has undergone significant improvements, the authors are encouraged to make the following adjustments:

1. Despite the authors' revisions, certain sections still require more practical discussion and improved coherence in the results discussion.

2. The discussions regarding both technical and practical aspects, as well as comparisons to recent studies on this topic, need thorough examination.

3. For effective comparison of the proposed scheme with the other schemes, the variables of the presented results should start from almost the same steady state. However, this is not the case in this paper.

4. The references require updating and upgrading, as many of them are outdated. Additionally, some references need to be more practical, impactful, comprehensive, and robust.

https://doi.org/10.1016/j.optcom.2020.126219

https://doi.org/10.1049/ote2.12111

I recommend accepting the paper after minor revisions.

7. PLOS authors have the option to publish the peer review history of their article (what does this mean? ). If published, this will include your full peer review and any attached files.

**Do you want your identity to be public for this peer review?** For information about this choice, including consent withdrawal, please see our Privacy Policy .

Reviewer #1: **Yes: ** Muhammad Zunnurain Hussain

Reviewer #3: **Yes: ** Ebrahim E. Elsayed

---

## [Author Response · Author response to Decision Letter 2]

20 Nov 2024

please find attach required reviewer responses

---

## [Decision Letter · Decision Letter 2]

26 Nov 2024

Optimization of Satellite-based Communication Links

PONE-D-24-33931R2

Dear Dr. Khan,

We’re pleased to inform you that your manuscript has been judged scientifically suitable for publication and will be formally accepted for publication once it meets all outstanding technical requirements.

Kind regards,

Elochukwu Ukwandu, PhD

Academic Editor

PLOS ONE

Additional Editor Comments (optional):

Reviewers' comments:

Reviewer's Responses to Questions

**Comments to the Author**

1. If the authors have adequately addressed your comments raised in a previous round of review and you feel that this manuscript is now acceptable for publication, you may indicate that here to bypass the “Comments to the Author” section, enter your conflict of interest statement in the “Confidential to Editor” section, and submit your "Accept" recommendation.

Reviewer #3: (No Response)

2. Is the manuscript technically sound, and do the data support the conclusions?

Reviewer #3: (No Response)

3. Has the statistical analysis been performed appropriately and rigorously? 

Reviewer #3: (No Response)

4. Have the authors made all data underlying the findings in their manuscript fully available?

Reviewer #3: (No Response)

5. Is the manuscript presented in an intelligible fashion and written in standard English?

Reviewer #3: (No Response)

6. Review Comments to the Author

Reviewer #3: The authors fully have addressed all the previous comments. The authors have made all the comments and corrections requested by the reviewers, so I recommend accepting the paper in the PLOS ONE journal.

7. PLOS authors have the option to publish the peer review history of their article (what does this mean? ). If published, this will include your full peer review and any attached files.

**Do you want your identity to be public for this peer review?** For information about this choice, including consent withdrawal, please see our Privacy Policy .

Reviewer #3: **Yes: ** Ebrahim E. Elsayed

---

## [Editor Report · Acceptance letter]

PONE-D-24-33931R2

PLOS ONE

Dear Dr. Khan,

I'm pleased to inform you that your manuscript has been deemed suitable for publication in PLOS ONE. Congratulations! Your manuscript is now being handed over to our production team.

Kind regards,

on behalf of

Dr. Elochukwu Ukwandu

Academic Editor

PLOS ONE